# Opportunities for Discovery Using Neoadjuvant Immune Checkpoint Blockade in Melanoma

**DOI:** 10.3390/ijms26062427

**Published:** 2025-03-08

**Authors:** Uma Nair, Emily Rakestraw, Georgia M. Beasley, Margaret H. O’Connor

**Affiliations:** Department of Surgery, Duke University, Durham, NC 27710, USA; uma.nair@duke.edu (U.N.); emily.rakestraw@duke.edu (E.R.); margaret.oconnor@duke.edu (M.H.O.)

**Keywords:** melanoma, lymph nodes, immunotherapy, neoadjuvant

## Abstract

Treatment of resectable advanced-stage melanoma with neoadjuvant immunotherapy is rapidly becoming the new standard of care due to significant improvements in event-free survival (EFS) compared to surgery first followed by immunotherapy. The level of responsiveness seen in patients receiving immune checkpoint inhibitors (ICIs) must be mechanistically understood not only for the standardization of treatment but also to advance the novel concept of personalized cancer immunotherapy. This review aims to elucidate markers of the tumor microenvironment (TME) and blood that can predict treatment outcome. Interestingly, the canonical proteins involved in the molecular interactions that immunotherapies aim to disrupt have not been consistent indicators of treatment response, which amplifies the necessity for further research on the predictive model. Other major discussions surrounding neoadjuvant therapy involve the higher-level investigation of ICI efficacy due to the ability to examine a post-treatment tumor molecularly and pathologically, which this review will also cover. As neoadjuvant ICI becomes the standard of care in advanced melanoma treatment, further research aiming to identify more predictive biomarkers of treatment response to advance medical decision-making and patient care should continue to be sought after.

## 1. Introduction

The historical approach to a patient with resectable advanced melanoma tumors was to remove the entire tumor first with surgery and then use immune checkpoint inhibitors (ICIs) to treat clinically occult disease, termed adjuvant therapy. However, neoadjuvant immune checkpoint inhibition, given prior to surgery for advanced melanoma (stage 3 or 4) amenable to surgical resection, should now be considered the standard of care after two large phase 3 randomized control trials (RCTs) demonstrated impressive, significant improvements in event-free survival (EFS) for patients receiving neoadjuvant therapy versus adjuvant therapy [1,2]. The main advantages of the neoadjuvant approach include treating clinically occult metastatic disease, the ability to assess tumor treatment effects, and the potential to limit the extent of surgery and, therefore, morbidity from surgery [1]. There is also a molecular basis for the use of neoadjuvant immune checkpoint blockade. There is mounting evidence that having abundant tumor antigens in place (not removing the clinically detected tumor) may augment anti-tumor responses to immune-directed therapies [2]. In patient analysis, neoadjuvant combination ipilimumab (anti-CTLA-4) and nivolumab (anti-PD-1) was associated with the expansion of more tumor-resident T-cell clones in the peripheral blood compared to adjuvant ipilimumab and nivolumab [3].

Although neoadjuvant immune therapy is now considered the standard of care in most cases for advanced melanoma amenable to surgical resection, the treatment still comes with the risk of serious side effects, and not all patients are expected to have tumor responses. Similar to patients with metastatic melanoma, predicting response or non-response to ICI prior to the initiation of therapy remains not standardized in the clinic due to the lack of consistent, high-performing biomarkers that can identify patients who will benefit from ICI, and many patients are exposed to side effects with no clinical benefit. Here, we will review the molecular features of melanoma indicative of response prior to the start of neoadjuvant ICI as well as the molecular features associated with favorable outcomes while receiving and after ICI therapy. After neoadjuvant therapy, the entire tumor is excised, which allows for the examination of the TME. This is not conducted or not possible in other metastatic melanoma patients, where only cross-sectional imaging is performed to assess response to ICI. Therefore, examining the TME post-ICI is unique to neoadjuvant therapy. We, therefore, have organized this review around the clinical concept of neoadjuvant therapy. First, we will explore the predictors of response before neoadjuvant ICI therapy, and then we will explore the predictors of response when the TME is studied after neoadjuvant therapy.

## 2. Pre-Treatment Predictive Molecular Markers

While neoadjuvant therapy has been shown to be superior with prolonged EFS, when compared to adjuvant therapy, not all patients respond to neoadjuvant therapy. Finding a biomarker to predict who will respond to neoadjuvant therapy or ICI therapy generally continues to be important for patients, but there is wide variation in clinical application. In melanoma, there is currently no widely or uniformly used biomarker to predict response. For patients with metastatic melanoma, we know that the response rates to anti-programmed death-ligand 1 (PD-1) are about 35% and that those to the combination of anti-PD-1 and anti-cytotoxic T-lymphocyte-associated antigen-4 (CTLA-4) are about 60% [4,5,6,7,8]. Because the majority of patients do not respond to anti-PD-1 monotherapy, and combination treatment comes with a high risk of adverse events, developing predictors of response could maximize patient benefit while minimizing toxicity.

Examining the tumor microenvironment (TME) is a critical first step in the evaluation of immunotherapy responsiveness [9]. The molecular TME of pre-treatment (baseline) tumor specimens has been vastly explored to determine if a favorable tumor response can be predicted before therapy starts and thereby minimize the risk of side effects or disease progression in patients who would not be predicted to respond. There are a wide range of factors that can predict ICI treatment outcomes. While no specific biomarker has been established for standard clinic practice, there are several common trends in the molecular features of the TME that have demonstrated importance in predicting response to ICI, which we will review.

### 2.1. Tumor-Infiltrating Lymphocytes

Effective cancer therapy promotes the priming and differentiation of naïve CD8^+^ T cells into antigen-specific cytotoxic T cells (CTLs), which leads to the elimination of tumor cells [10,11]. ICIs target inhibitory receptors on CD8^+^ T cells, which potentiate effector antitumor CD8^+^ T-cell function [12]. At the inception of ICI therapy and early in the study of the melanoma TME, it was noted that the presence of tumor-resident immune cells was paramount to the successful treatment of the tumor. The presence of high levels of these tumor-infiltrating lymphocytes (TILs) has been linked to a favorable prognosis [13,14,15,16].

Certain specific subsets of TILs have been linked to ICI responsiveness. The first of these subsets is CTLA-4- and PD-1-expressing CD8^+^ T-cell presence compared to the non-responder phenotype clearly showing the presence of CD8^+^ TILs that lacked PD-1/CTLA-4 expression [17]. Another subset of TILs associated with favorable responses to ICI are CD103^+^CD39^+^CD8^+^ TILs that show an exhausted tissue-resident memory phenotype and have a distinct T-cell receptor repertoire [18,19]. Further, CD69^+^CD103^+^ tumor-resident CD8^+^ T cells were associated with improved melanoma-specific survival in immunotherapy-naïve melanoma patients [20], and, finally, CD39^+^CD8^+^ tumor-resident T cells showed weaker but improved clinical outcomes with ICI therapy [21]. Additionally, tumoral B lymphocytes have also been explored. The presence of tertiary lymphoid structures, comprised of a CD3^+^ T-cell-rich region, a CD20^+^ B-cell-rich region, plasma cells (PCs), dendritic cells (DCs), and fibroblastic reticular cells (FRCs), in pre-treatment tumors has been reported to predict response to ICI therapy [22]. Further, Cabrita et al. derived a gene signature associated with tertiary lymphoid structures that predicted clinical outcomes in patients treated with immune checkpoint blockade [23].

Of note, it has been assumed that PD-1 blockade works by reinvigorating these pre-existing exhausted TILs. However, other work suggested that PD-1 blockade recruits new effector T cells into the TME [15]. Thus, the presence of TILs pre-treatment may demonstrate intact innate immune function or intact ability to traffic T cells to the tumor. Although ICIs can renew the effector function of CD8^+^ T cells within the tumor or recruit new effector cells, the majority of patients with metastatic melanoma still fail to respond to checkpoint blockade [9,15]. Accordingly, the presence of TILs is currently not used in clinical practice to predict responses to ICI therapy.

### 2.2. Immune Evasion of PD-1/PD-L1 Axis

The PD-1/PD-L1 axis is essential for suppressing the continuous activation and proliferation of different immune effector cells. When PD-1 engages its ligands, it induces a state of T-cell dysfunction via exhaustion [13]. PD-L1 on antigen-presenting cells (APCs) can control regulatory T-cell differentiation and immunosuppressive activity. PD-L1 expression may be either constitutive when expressed at a low level, for example, on resting lymphocytes or APCs, or inducible when its expression is upregulated by inflammatory events [24,25,26]. In a complementary role to PD-L1 function in regulating T-cell activity is CTLA-4, another critical immune checkpoint receptor that exerts regulatory effects on T-cell activation and is constitutively expressed on regulatory T cells or upregulated on conventional T cells after activation [13]. These two immune checkpoint pathways are targeted by ICI as they become dysregulated in the TME.

More specifically, in the melanoma TME, the immune system becomes functionally exhausted after prolonged exposure to melanoma antigens and the chronic inflammatory state. This occurs due to the overactivation of inhibitory checkpoints on immune cells (PD-1/PD-L1, CTLA-4, LAG3, etc.), chronic exposure to inflammatory cytokines, and a negative feedback loop for cytotoxic T cells [27]. Simultaneously, the TME employs additional mechanisms that perpetuate a poor cytotoxic phenotype in T cells, including enrichment for tumor-associated macrophages (TAMs), regulatory T cells (T regs), and myeloid-derived suppressor cells (MDSCs). The overproduction of immunosuppressive cytokines (IL-10, TGFβ1, and TGFβ2) and the enzyme indoleamine 2,3-dioxygenase (IDO) and the downregulation of both class I and class II major histocompatibility complex (MHC) antigens are the primary causes of the ineffective killing of tumor cells [27].

It was originally hypothesized that PD-L1 expression (confirmed with immunohistochemistry staining for PD-L1) was essential for a melanoma patient to respond to ICI therapy. It then became clear through many clinical trials in melanoma that PD-L1 tumor expression does not reliably predict response to therapy. In a review of 451 patients, PD-L1 expression in pre-treatment tumor biopsy samples was correlated with response rate, progression-free survival, and overall survival; however, a subset of patients with PD-L1–negative tumors also achieved durable responses [28]. Another study showed that the combination of nivolumab plus ipilimumab in previously untreated patients with metastatic melanoma yielded a response rate of 72% among patients who were PD-L1-positive and 55% among patients who were PD-L1-negative [29]. Several researchers have shown that the correlation between response in melanoma (and other malignancies) and PD-L1 expression is likely due to the detection of PD-L1 on infiltrating immune cells rather than on neoplastic cells [30,31]. It was shown that macrophages and stromal cells expressed PD-L1 during the progression/relapse phase [32].

The mechanisms behind this dichotomy between PD-L1 expression and responsiveness to anti-PD-1 therapy despite low PD-L1 expression are highly complex. It has been theorized that alternative immune pathways in PD-L1-negative tumors are active, leading to the tumor still being responsive to ICI therapy. First, PD-L1 expression on tumor cells was predominantly observed in spatial association with CD8^+^ T cells, consistent with an adaptive mechanism of expression [33]. Further, some tumor microenvironments can induce PD-L1 on Natural Killer (NK) cells via AKT signaling, resulting in enhanced NK-cell function and preventing cell exhaustion. Anti-PD-L1 monoclonal antibodies directly act on PD-L1-positive NK cells, which then target PD-L1-negative tumors via a p38 pathway [34].

The Vilain group found that pre-treatment intratumoral and peritumoral PD-1^+^ T-cell densities were sevenfold and fivefold higher, respectively, in responders compared with non-responders and correlated with the degree of radiologic tumor response [35]. The pre-treatment PD-L1 expression on tumors and macrophages was not significantly different between the patient groups, but tumoral PD-L1 and macrophage PD-L1 expression was higher in the early-treatment tumors (within 2 months of treatment) of responders compared to non-responders. Responder early-treatment biopsies (compared with pre-treatment) also showed significant increases in intratumoral CD8^+^ lymphocytes and CD68^+^ macrophages [35].

Furthermore, there has been great debate in the field over standardizing the immunohistochemistry staining and subsequent scoring of PD-L1 expression in melanoma tumors. Several scoring systems have been employed, including the tumor proportion score (TPS) [36], the combined positive score (CPS) [36,37], and the melanoma (MEL) score [28,36]. The utilization of different scoring systems, different PD-L1 antibody clones and the fact that the optimal “cut-off” level remains ill-defined represent a matter of debate in practice. The 28-8 clone (Dako) was employed in the CheckMate067 study, a phase III trial involving patients with treatment-naïve advanced melanoma [5,38]. In that study, tumor PD-L1 expression alone was not predictive of efficacy outcomes, and no significant relationship between PD-L1 expression and improvement in progression-free survival (PFS) in patients receiving combined immunotherapy was found. A different PD-L1 testing antibody was used in the Keynote006 trial [39], which used the 22C3 clone (Dako). In that study, they reported that two-year survival rates were higher with pembrolizumab monotherapy compared to ipilimumab in patients with PD-L1-positive tumors (PFS 33.2% versus 13.1%; OS 58.4% versus 45.0%) and in PD-L1-negative tumors (PFS 14.9% versus non-responder; OS 43.6% versus 31.8%) [40]. These trials each reported differences in PD-L1 expression and, thus, responsiveness; however, different clones and blinding studies were employed. This example highlights one such instance of the issues with variability and difficulty in drawing conclusions between responsiveness to anti-PD-1 therapy and PD-L1 expression by the tumor. Therefore, PD-L1 expression is currently not used to guide treatment decisions. Furthermore, PD-L1 expression was not required for entry in the OpACIN trial nor for other neoadjuvant immune therapy trials in melanoma. This necessitates finding other ways to accurately predict patients who will or will not respond to ICI therapy.

### 2.3. Gene Expression Profiles

Due to the inability to reliably utilize PD-L1 expression alone, many groups have moved to identify other biomarkers that can be used clinically to predict responders/non-responders to ICI. Several groups have come up with gene signature pattern predictive models with varying clinical correlation success.

Jiang et al. developed a Tumor Immune Dysfunction and Exclusion (TIDE) gene signature, which could accurately predict cancer immunotherapy response. It is mechanistically based on the two primary mechanisms of tumor immune evasion: induction of dysfunctional T cells by tumors that have high levels of cytotoxic T lymphocytes (CTLs) and prevention of T-cell infiltration in tumors with low levels of CTLs. Their data were validated with a false discovery rate set to <0.1 and rigorously tested across five large-scale cancer cohorts (TGCA, PRECOG, etc.) [10]. Unfortunately, the TIDE signature consists of 770 genes, making it difficult for widespread clinical application. Another recent study calculated the immune score of melanoma samples by using the ESTIMATE (Estimation of STromal and Immune cells in MAlignant Tumor tissues using Expression data) algorithm, and 25 genes that best correlated with the immune score were identified [41]. However, there are several limitations with this model. First, prognosis-related genes were not taken into consideration. Second, no statistical models were used to avoid overfitting. Finally, and most importantly, this 25-gene signature showed no significant association with response to immune checkpoint blockade therapies [41].

The Lo group used whole exome sequencing of pre-treatment melanoma tumors for a gene signature linked to non-responders. They found that innately resistant tumors to anti-PD-1 therapy display a transcriptional signature (referred to as the IPRES or Innate anti-PD-1 Resistance) indicating the concurrent upregulation of genes involved in controlling mesenchymal transition, cell adhesion, ECM remodeling, angiogenesis, and wound-healing [42]. Their findings were validated with a multiple hypothesis correction of FDR *p* < 0.25 and also rigorously tested via TGCA, and the TIDE signature was as well. They identified that acquired resistance to MAPK-targeted therapy has been correlated with the depletion of intra-tumoral T cells, exhaustion of CD8^+^ T cells, and loss of antigen presentation [43]. Their findings show that those patients at risk of failure with anti-PD-1 treatment have high mutational burden and altered MAPK pathways, suggesting co-treatment with anti-PD-1 and MAPK inhibitors.

Further work sequencing pre-treatment, during-treatment, and treatment-resistant tumors is needed to narrow down the number of genes needed in the predictive model and track how TMEs change over time/with treatment to make it more specific and accessible for widespread use.

### 2.4. Combination Strategies

Given that there remains no clear predictive strategy that can reliably and reproducibly determine patients who will be responders or non-responders to ICI, combinations of multiple predictive strategies should be utilized. Tumor mutational burden (TMB) and the T-cell–inflamed gene expression profile (GEP) are emerging predictive biomarkers for anti-PD-1 therapy [9]. Both PD-L1 expression and GEP are inflammatory biomarkers indicative of a T-cell–inflamed tumor microenvironment, whereas TMB and high microsatellite instability (MSI-H) are indirect measures of tumor antigenicity generated by somatic tumor mutations. The T-cell–inflamed GEP contains IFN-γ-responsive genes related to antigen presentation, chemokine expression, cytotoxic activity, and adaptive immune resistance, and these pathways were necessary, but not always sufficient, to see clinical benefit from ICI treatment [44]. The Ayers group then clinically validated biomarkers predictive of response to the anti-PD-1 monoclonal antibody pembrolizumab, including PD-L1 expression and MSI-H. They found that objective response rates were the strongest in patients with high GEPs and high TMB, moderate in those with high expression of one of the two, and reduced or absent in those with GEP- and TMB-low tumors. Additionally, longer progression-free survival times were seen in patients with higher levels of the GEP signature in melanoma [9]. Thus, these two complimentary, but not overlapping, pathways can be used in combination to more accurately predict responsiveness to ICI. High or low TMB can most likely also be combined with any of the gene expression profile sets described here (GEP, TIDE, or IPRES) to lead to a higher ability to reliably predict patients who will be responders or non-responders to ICI.

This was further explored and validated in a mouse model of melanoma by the Perez-Guijarro group, which found that TIL densities after CTLA-4 blockade correlated better with efficacy than did those at baseline. They also found that TIL density alone is not a sufficient predictor of resistant tumors and that one must also take into account the functionality of the T cells [45]. CD8^+^ and CD4^+^ T-cell dysfunction profiles sustained by the intratumoral macrophages and dendritic cells (DCs) explained the resistance in their mouse model to anti-CTLA-4 therapy despite high tumor mutational burden (TMB) and TIL quantities. By contrast, the enrichment of tumor-promoting macrophages, reduced NK cells, and high T regs supported the low-TIL and ‘cold’ TME of resistant tumors. Notably, they found a similar distribution of these immune cells in non-responder patients to ICI. Moreover, they applied the TIDE gene signature profile to the mouse models and obtained consistent predictions of response [45].

Furthermore, the Long group showed that high TMB, neoantigen load, expression of IFN-γ-related genes, programmed death-ligand expression, presence of T cells in the tumor microenvironment, and low *PSMB8* methylation (and, thus, high expression), are associated with responders to immunotherapy [46]. These studies highlight that the utilization of combinations of the current predictive strategies can allow us to more precisely determine patient response rates.

There has been much work dedicated to predicted response to ICI in pre-treatment tissue, yet no standard assay is used in the clinic. Response to ICI is complex and likely dependent on many factors, as outlined. Further work should include these predictors in prospective studies.

## 3. Post-Treatment Predictive Pathologic and Molecular Markers

### 3.1. Pathologic Response

For patients with metastatic melanoma receiving ICI therapy, disease assessments are performed with interval cross-sectional imaging (e.g., CT scan). Often, treatment “response” is determined by whether tumors are decreasing in size, stable, or increasing in size and number. However, particularly with ICI therapy, imaging is not always accurate, and more precise estimates of treatment effects on tumors are needed [47]. With neoadjuvant therapy, pathologic examination of specimens at the time of removal after ICI can enable real-time assessment of treatment effects. Indeed, evaluation of post-treatment TME can assist in prognosis and in further decision-making regarding additional therapies. Not surprisingly, there is some overlap in markers between the pre- and post-treatment TME, but there are aspects of the post-treatment TME that are unique.

Pathological response can be determined by microscopically examining the viability of tumor cells following surgical removal after ICI therapy. In patients with breast cancer and non-small-cell lung carcinoma, where neoadjuvant therapy is also increasingly being used, standardized pathological evaluation has proven important in prognostication and personalized treatment. In melanoma, criteria have been developed to standardize the assessment of tumors after ICI therapy. Tetzlaff et al. used criteria from other cancers and approached the pathological assessment of the melanoma “tumor bed” with a compositional analysis of the resected lymph node in melanoma patients, as seen below [48]:% viable tumor + % tumoral melanosis/necrosis + % fibrosis/fibroinflammatory stroma = 100%

Pathologists can then grade pathological response based on the % viable tumor:Complete pathologic response (pCR), no viable tumorMajor pathologic response (MPR), ≤10% viable tumorPathologic partial response (pPR), >10 to ≤50% viable tumorPathologic non-response (pNR), >50% viable tumor

Indeed, pathologic response after ICI neoadjuvant treatment is the most powerful predictor of EFS [49]. In the NADINA clinical trial studying neoadjuvant ipilimumab plus nivolumab followed by surgery, the estimated 12-month recurrence-free survival was 95.1% in patients in the neoadjuvant group who had a major pathological response, 76.1% among those with a partial response, and 57.0% among those with a non-response as shown in Table 1 [50]. Thus, pathologic response is a highly prognostic and robust marker of response to ICI therapy.

Furthermore, pathologic assessment of tumor after neoadjuvant ICI treatment is essential to identify patients at risk for progressive disease and can be utilized to curate personalized treatment plans such as for those patients who would benefit from adjuvant treatment.

Patients with non-response on pathologic examination after ICI can be changed to alternative therapies or undergo treatment escalation. In the NADINA trial, patients with BRAF mutations who were only partial or non-responders after ICI were initiated on adjuvant dabrafenib plus trametinib (BRAFi/MEKi). Further, in NADINA, patients with major pathologic responses received no additional therapy, with seemingly no detriment to clinical outcome. This represents a major advancement in reducing cost and patient exposure to unnecessary treatments and adverse side effects. In another ongoing study (NCT04013854), the pathologic response after neoadjuvant nivolumab was used to stratify adjuvant therapy as subjects with less than a PathCR/nearCR were randomized 1:2 to either adjuvant nivolumab or adjuvant ipilimumab plus nivolumab.

Finally, specific features of the pathologic response may be more predictive of clinical outcomes. After neoadjuvant ICI as part of the OpACIN-neo trial, the immunotherapeutic response subtype with high fibrosis (in the specimen) had the strongest association with lack of recurrence (*p* = 0.008) and prolonged recurrent-free survival (RFS) (*p* = 0.019) [51]. Further, the number of B lymphocytes was significantly increased in patients with the high fibrosis subtype treatment response (*p* = 0.046) [51]. In a separate study, tumor necrosis after neoadjuvant anti-PD-1 therapy in combination with increased tumor-infiltrating lymphocytes in necrotic tumor necrosis (nTIL) correlated with major pathologic response and increased 5-year recurrence-free survival [52].

### 3.2. T-Cell Intratumoral Infiltration and T-Cell Receptor (TCR) Clones

In addition to standard pathologic assessment, there may be other molecular markers of response to ICI, including T-cell intratumoral infiltration and expansion of TCR clones. The Kirkwood group analyzed tumor samples before and after ipilimumab neoadjuvant therapy. There was a significant increase in circulating regulatory T cells (T regs; CD4^+^CD25hi^+^Foxp3^+^) that was also associated with improved PFS (*p* = 0.034). In the tumor collected post-surgery, there was a significant increase in CD8^+^ T cells (*p* = 0.02). Ipilimumab also induced increased tumor infiltration by fully activated (CD69^+^) CD3^+^/CD4^+^ and CD3^+^/CD8^+^ T cells, with evidence of induction/potentiation of memory T cells (CD45RO^+^) [53].

Further work examined the effects of neoadjuvant anti-PD-1 therapy. Exhausted CD8^+^ T cells (T_EX_) have shown signs of reinvigoration only 7 days after a single dose of anti-PD-1 [54]. In patients receiving a single dose of neoadjuvant ICI, there was a marked increase in Ki67^+^ CD8^+^ T cells at day 7 after initiation [54]. Immunofluorescence staining of tumor samples pre- and post-treatment revealed increased CD8^+^ T-cell infiltration, with the majority of these cells bound to pembrolizumab and displaying an exhausted phenotype post-treatment [54]. Despite an increase in the proportion of PD-1^+^CTLA-4^+^ CD8^+^ T cells in the tumor 3 weeks post-treatment, this cell population did not have increased Ki67 and was, thus, not expanding. Further, there was no expansion of PD-1^+^CTLA-4^+^ or PD-1^+^ CD8^+^ T cells in the peripheral blood [54]. Together, these findings by Huang et al. [54] may indicate either that tumor immune reactivation is early and not sustained or that T cells are reinvigorated in the periphery prior to being trafficked to the tumor. The rapid response to anti-PD-1 supports the hypothesis that there is an existing intratumoral T-cell population that becomes revitalized with the therapy.

Further work comes from patients receiving combination neoadjuvant immune therapy. In the OpACIN trial, ipilimumab 3 mg/kg + nivolumab 1 mg/kg therapy was given to 10 patients neoadjuvantly and 10 patients adjuvantly for comparison [3]. In the neoadjuvant group, 7/9 evaluated patients experienced a pathologic response after two courses of therapy. Patients in the neoadjuvant arm of the study were found to have increased T-cell clonal expansion [3]. Interestingly, TCR clonal expansion has been proven to be important in the response to anti-PD-1 therapy in other malignancies. Neoadjuvant anti-PD-1 therapy has also emerged as a prominent treatment for non-small-cell lung cancer (NSCLC) [55]. Zhang et al. enrolled 21 patients with resectable NSCLC to receive two doses of neoadjuvant nivolumab and performed serial TCR sequencing to detect patterns of T-cell clones in the TME that had been expanded in the periphery. In trends similar to those seen in melanoma patients, they found that the top 1% of intratumoral T-cell clones in major pathological responders were primarily peripheral clones that trafficked to the tumor site [55]. These findings support the hypothesis that a pre-existing T-cell population is rejuvenated and expanded in the periphery upon initiation of immune checkpoint blockade inhibitors and then recruited into the TME [56].

### 3.3. B Cells, Tertiary Lymphoid Structures

Additionally, B cells and tertiary lymphoid structures (TLSs) have been identified as possible markers of response to ICI in post-treatment tissue and blood [57]. Similar to the identification of TCR clones, B-cell changes can be detected in the peripheral blood and tumor following initiation of ICI. Early biomarker changes in response to ICI can aid in post-surgery treatment decisions and help to avoid exposing unresponsive individuals to unnecessary toxicities following surgery. In a randomized phase 2 study (NCT02519322), 23 patients with resectable melanoma underwent either neoadjuvant nivolumab or combined ipilimumab with nivolumab then surgical resection followed by adjuvant nivolumab [58]. Immune profiling via immunohistochemistry revealed the increased expression of lymphoid markers, including CD20 (B cells), in between baseline and early-treatment tumor samples for responders versus non-responders, with the early-treatment samples being a stronger predictor of response than the baseline samples [58].

Using samples from the same cohort of patients from the NCT02519322 study, Helmink et al. analyzed the potential role of B cells and tertiary lymphoid structures in the response to ICB treatment. The density of CD20^+^ B cells and TLSs was higher among patients who responded to neoadjuvant therapy versus non-responders in early-/post-treatment analysis, but there was no significant difference in the baseline (pre-treatment) samples [57]. Further, these B cells were found to be localized in the TLSs of tumors in the patients who were responding to therapy. So, while TLS pre-treatment may be predictive of response to ICI as discussed in the pre-treatment section, the formation of TLSs on or after treatment may also be a powerful prognostic marker of treatment response.

## 4. Conclusions

In this review, we have highlighted molecular predictors of response to neoadjuvant ICI pre-treatment and post-treatment. Pre-treatment biomarkers can prevent exposure to side effects for patients who will likely not benefit from treatment. These are best utilized in combination, as no one biomarker has been predictive of responsiveness yet. Post-treatment biomarkers can assist in medical decision-making for additional therapies that may be needed and surveillance plans. Neoadjuvant therapy allows for thorough examination of the tumor microenvironment post-ICI therapy and is now considered the standard of care for patients with advanced melanoma amenable to surgical resection. Importantly, for any study of the TME, tumor heterogeneity must be taken into account and might be a confounding factor when treatment decisions are made based on tumor biopsies or only examining a section of the tumor. Even when the melanoma tumor left in situ during neoadjuvant treatment is analyzed, relapse might be due to small numbers of surviving cell clones that can be overlooked. It could be important to focus on the surviving melanoma clones and characterize them with molecular techniques in order to define better prognostic features.

Importantly, there is not a biomarker of response to ICI therapy that is routinely used in the clinic. Future clinical trials should incorporate biomarkers so that we can advance the field and better select patients for therapies. Neoadjuvant therapy is associated with improved clinical outcomes for patients; further exploration of the molecular basis for tumor response is possible and can provide extreme value for medical decision-making.

## Figures and Tables

**Table 1 ijms-26-02427-t001:** Summary of NADINA trial.

	Neoadjuvant Group (n = 212)	Adjuvant Group (n = 211)
Design	Neoadjuvant therapy followed by surgery	Surgery followed by adjuvant therapy
	IPI + NIVO, 2 cycles	NIVO, 12 cycles
EFS at 10-month follow-up (% patients)	83.7	57.2
MPR/EFS 12 months (% patients) pPR/EFS 12 months (% patients) pNR/EFS 12 month (% patients)	59/95.1	
	8/76.1	N/A
	26/57	

## Data Availability

The authors permit the sharing of all the data.

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
