# Peer review of "Opportunities for Discovery Using Neoadjuvant Immune Checkpoint Blockade in Melanoma"

_ijms, 2025, doi:10.3390/ijms26062427_

Round 1
Reviewer 1 Report
Comments and Suggestions for Authors
In this review article, Nair et al aim to comprehensively describe the molecular predictors of response to neoadjuvant ICI pre-treatment and post-treatment. They emphasize that there is an important need for accurate biomarker tests to correctly select melanoma patients for neoadjuvant therapy while sparing likely non-responders the toxicities associated with ICI. They also review the post treatment histologic features and immune cell subsets that correlate with poor response.
While this is an important area of research and this review highlights a lot of good data, it is challenging to read and could be better organized. Furthermore, the title is a bit misleading as besides the sections on tumor PD-L1 expression and gene signature tests, there isn't much discussion regarding genomic alterations and specific mutations that are associated with ICI response. A significant portion of the review is focused on pathologic response and the presence of T cell/B Cell subsets. Additionally, there is overlap between the pretreatment and post treatment sections that adds to the confusion of the review- for example the presence of tertiary lymphoid structures in baseline melanoma tumors has been associated with improved outcomes and ICI response, but TLS are only described in the post treatment section. I believe that the review could be significantly improved with clearer organization and reformatting.
Some other comments:
- At the end of the introduction, the authors write “predicting response or non-response to ICI prior to initiation of therapy remains not standardized in the clinic”. They need to emphasize that testing is not standardized because a good test does not currently exist.
- In first paragraph of pre-treatment predictive molecular markers section, they state that response rate to combination anti-PD-1 and CTLA-4 is 70%. This numbers seems too high as the CheckMate-067 founds a response rate of 58%, and one of the referenced studies for their response data is a breast cancer study not a melanoma study.
- While I agree with the authors that the presence of TILs is not enough to predict response (as TILs may include non-tumor reactive bystander T cells), there are numerous studies that are in concordance with one another and demonstrate that the presence of particular subsets of TILs do predict response and are prognostic. These subsets include: PD-1+ CTLA-4+ CD8+ TILs (PMID 27525433), CD103+ and/or CD39+ TILs (PMID: 30006565, PMID: 29599411, PMID: 36574773), tissue-resident memory (TRM) CD8+ TILs (PMID: 33205076), and tumor-reactive CD8+ TILs. While these subpopulations are not routinely measured in the clinical setting to predict immunotherapy response, the authors should include a brief discussion of these studies. They also allude to the predictive features of TIL subsets in other sections of the review which goes against their statement that TILs do not predict response.
- Additional information regarding test metrics (PPV/NPV) to describe the accuracy of the different GEP test should be added.
- The strong association between the presence of TLS in baseline tumor with ICI response
- The Molecular patterns of non-responses section seems redundant without contributing much to the manuscript
Comments on the Quality of English Language
The review was hard to read at times which I believe could be improved with increased focus on improving the grammar/screening for typos and organizational flow. The English is mostly fine but could be written more concisely and clearer.
Reviewer 2 Report
Comments and Suggestions for Authors
The manuscript provides a well balanced overview of molecular and biological markers that influence response towards neoadjuvant melanoma treatment.
I have some minor suggestions:
- The authors could discuss previous work of the Kirkwood group (PMID: 24498358).
- The authors could discuss if adding a vaccine to a neoadjuvant treatment might enhance or reduce treatment response. It might depend on whether a vaccine is given prior or after neoadjuvant treatment.
- The authors might discuss the problem of tumor heterogeneity which is greatest in the primary tumor and might be a confounding factor when treatment decisions are made based on tumor biopsies. Even when the whole remaining melanoma is analyzed during neoadjuvant treatment, relapse might be due to small surviving cell clones that might be overlooked or not taken into account. It could be important to focus on the surviving melanoma clones and characterize them with molecular techniques in order to define better prognostic features.
Round 2
Reviewer 1 Report
Comments and Suggestions for Authors
Thank you for submitting the revised manuscript- the improved organization makes the manuscript much easily to follow and the added content adds to its comprehensiveness.
My only comment is that it's unclear to me what the added paragraph on RNA vaccines adds to the manuscript as the article discusses its therapeutic value but not its predictive biomarker potential. Clarification on how this paragraph relates to the topic of the manuscript is suggested (or the paragraph's removal).
Author Response
Reviewer comment: Thank you for submitting the revised manuscript- the improved organization makes the manuscript much easily to follow and the added content adds to its comprehensiveness.
My only comment is that it's unclear to me what the added paragraph on RNA vaccines adds to the manuscript as the article discusses its therapeutic value but not its predictive biomarker potential. Clarification on how this paragraph relates to the topic of the manuscript is suggested (or the paragraph's removal).
Response: We agree and had not included information on vaccines since focus was on neoadjuvant systemic immune therapies and predictive potential for ICI responses. However, Reviewer 2 had recommended we add commentary on vaccines and so we added. We are happy to follow guidance from reviewer 1 and agree would make sense to delete it, but then worry reviewer 2 will want vaccines added in. Please advise. For now we plan to have the paragraph deleted.